Citation: *Molecular Systems Biology* 9:663
www.molecularsystemsbiology.com

# Systematic identification of proteins that elicit drug side effects

**Michael Kuhn**[1,4], **Mumna Al Banchaabouchi**[2,5], **Monica Campillos**[1,6], **Lars Juhl Jensen**[1,7], **Cornelius Gross**[2], **Anne-Claude Gavin**[1] **and Peer Bork**[1,3,*]

[1] Structural and Computational Biology Unit, European Molecular Biology Laboratory, Heidelberg, Germany, [2] Mouse Biology Unit, European Molecular Biology Laboratory, Monterotondo, Italy and [3] Max-Delbrück-Centre for Molecular Medicine, Berlin, Germany
[4] Present address: Biotechnology Center, TU Dresden, 01062 Dresden, Germany
[5] Present address: Preclinical Phenotyping Facility, Campus Science Support Facilities GmbH, Dr Bohr Gasse 3, 1030 Vienna, Austria
[6] Present address: Institute for Bioinformatics and Systems Biology (MIPS), Helmholtz Center Munich—German Research Center for Environmental Health (GmbH), Ingolstädter Landstraße 1, 85764 Neuherberg, Germany
[7] Present address: Novo Nordisk Foundation Center for Protein Research, Faculty of Health Sciences, University of Copenhagen, Blegdamsvej 3b, 2200 Copenhagen, Denmark
* Corresponding author. Structural and Computational Biology Unit, European Molecular Biology Laboratory, Meyerhofstrasse 1, Heidelberg 69117, Germany. Tel.: + 49 6221 387 8526; Fax: + 49 6221 387 517; E-mail: bork@embl.de

Side effect similarities of drugs have recently been employed to predict new drug targets, and networks of side effects and targets have been used to better understand the mechanism of action of drugs. Here, we report a large-scale analysis to systematically predict and characterize proteins that cause drug side effects. We integrated phenotypic data obtained during clinical trials with known drug–target relations to identify overrepresented protein–side effect combinations. Using independent data, we confirm that most of these overrepresentations point to proteins which, when perturbed, cause side effects. Of 1428 side effects studied, 732 were predicted to be predominantly caused by individual proteins, at least 137 of them backed by existing pharmacological or phenotypic data. We prove this concept *in vivo* by confirming our prediction that activation of the serotonin 7 receptor (HTR7) is responsible for hyperesthesia in mice, which, in turn, can be prevented by a drug that selectively inhibits HTR7. Taken together, we show that a large fraction of complex drug side effects are mediated by individual proteins and create a reference for such relations.
*Molecular Systems Biology* **9:** 663; published online 30 April 2013; doi:10.1038/msb.2013.10
*Subject Categories:* computational methods; molecular biology of disease
*Keywords:* computational biology; drug targets; side effects

## Introduction

While the aim of therapeutic pharmacological treatment is to restore the function of protein interaction networks that have been disturbed by disease, drugs often induce unintended changes in the body causing debilitating side effects that may require additional therapy or even lead to discontinuation of the drug. Specific protein targets have been recognized not only as important for the therapeutic effect of drugs, but also for the occurrence of side effects (Whitebread *et al*, 2005; Blagg, 2006; Imming *et al*, 2006; Tatonetti *et al*, 2009; Berger and Iyengar, 2011). In turn, perturbations induced at individual proteins influence pathways and functional modules, eventually leading to an observable phenotype (Scheiber *et al*, 2009). A number of studies have successfully used existing biochemical data to identify candidate targets for particular side effects. However, most systematic surveys toward proteins that constitute the molecular basis of side effects have concentrated on a small number of proteins or side effects (seeBender *et al*, 2007; Yang *et al*, 2009, 2010; Berger and

Iyengar, 2011; and Supplementary information). For example, the causes of several cardiovascular side effects were identified in a survey of the literature (Whitebread *et al*, 2005). Potential side effects of drug candidates were proposed to be predictable from *in vitro* protein binding without investigating a causal connection between protein binding and side effects (Krejsa *et al*, 2003). There have also been a number of efforts to extend the drug–target network to help explaining side effects (Xie *et al*, 2007, 2009; Lounkine *et al*, 2012). These studies employed various methods to predict new drug targets, but implicitly assumed a causal connection between these novel drug targets and side effects. Other studies of side effects imply that a systematic association with proteins is feasible. For example, pathways perturbed by drugs were related to the occurrence of side effects (Scheiber *et al*, 2009), and a method was proposed to find clusters of related drugs, targets and side effects (Mizutani *et al*, 2012). Again, no global benchmark was performed to show that the clusters correspond to causal relations. We have shown earlier that shared side effects between drugs can be used to predict shared targets

(Campillos *et al*, 2008), with the underlying assumption that drug targets are associated with specific patterns of side effects independent of the drug that binds the protein. Here, we integrated drug–target and drug–side effect relations to identify target proteins that elicit specific side effects. By recording whether agonistic or antagonistic changes of the targets cause the unwanted side effect, we can also propose ways by which the side effect can be counter-acted and confirm this concept in a mouse model. In contrast to the previous studies mentioned above, we based our analysis on the complete set of drug–target and drug–side effect relations and performed various benchmarks against the independent data and literature. To show the predictive value of our approach, we also tested a predicted side effect–target relation using a mouse model.

## Results

### Detection of overrepresented protein–side effect pairs

To systematically identify drug targets that cause a particular side effect, we combined side effect data for approved therapeutic drugs from SIDER 2 (Kuhn *et al*, 2010) with drug–target binding data from multiple sources (see Materials and methods) (Roth *et al*, 2000; Imming *et al*, 2006; Okuno *et al*, 2006; Günther *et al*, 2008; Wishart *et al*, 2008; Flockhart, 2009; Gaulton *et al*, 2011) as stored in the STITCH 3 database (Kuhn *et al*, 2012). Importantly, we also included information about whether the drug acts as an agonist or antagonist (or, for enzymes, as activator or inhibitor), as this information is often critical to predict the physiological mechanism of the side effect. Our initial data set contains annotations for 841 drugs and 1465 human targets and off-targets. After removing redundant data as well as target proteins and side effects that are associated with less than five drugs (for which we cannot make confident predictions, see Supplementary Figure 1), we arrived at a combined network of 1428 side effects, 550 marketed drugs and 296 drug targets (see Supplementary Figure 2 for histograms). Next, we predicted causal relationships between protein binding and side effects by searching for statistically significant correlations between the 5579 drug–target binding relations and 57 388 drug–side effect relations in our data set (Figure 1). These correlations allowed us to determine the set of drugs that bind a given target and elicit a particular side effect. We then calculated the significance (by *P*-value using Fisher's exact test) of overrepresentation against the background incidence of the respective side effect. The *P*-values are then adjusted to control for multiple hypothesis testing, yielding *q*-values (as quantified by the 'qvalue' package for the R programming language; Storey and Tibshirani, 2003). More formally, *q*-values represent the minimum false discovery rate for which the association will be regarded as significant.

Most drugs bind to sets of pharmacologically similar proteins, for example, members of the same protein family or complex (Kuhn *et al*, 2008; Hopkins, 2008). While it is likely that only one of the targets is responsible for a given side effect, a number of related targets will be predicted to cause the side effect in such cases. Thus, counting individual proteins

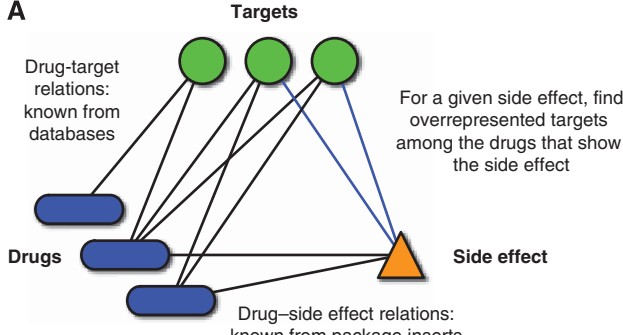

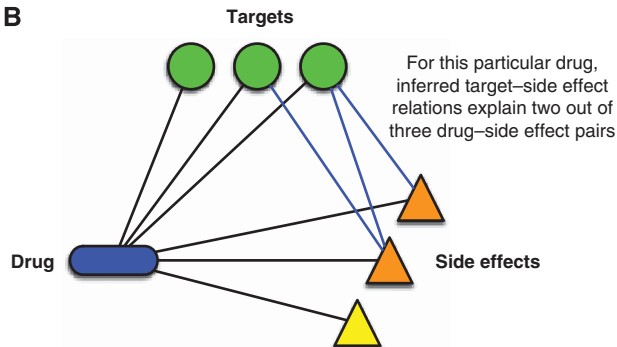

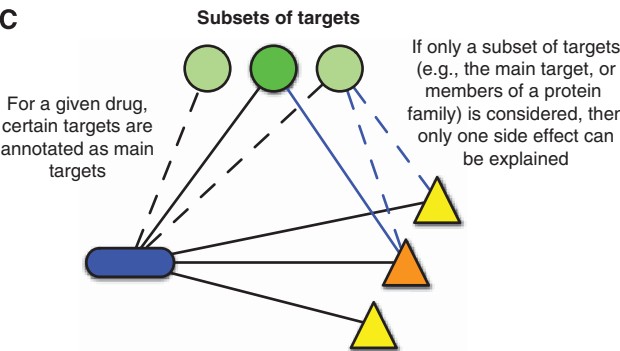

**Figure 1** Concept. (**A**) By combining known drug–target and drug–side effect relations, overrepresented protein–side effect pairs are identified. Benchmarking verifies that the observed correlations reflect causal relations (see Figures 2 and 3). (**B**) Once protein–side effect relations have been found, they can be used to explain actual side effects observed for drugs by looking for proteins that are predicted to cause the side effect among the drug's targets. (**C**) To identify the contributions of certain target classes (e.g., main targets or GPCRs), we filter the drug targets accordingly. Then, only side effects explained by proteins that belong to this subset of targets are considered. For example, main targets are known for many drugs. We consider these to be the targets traditionally associated with the therapeutic mechanism of action (see Materials and methods). A given target can be a main target for one drug, but an off-target for other drugs. Therefore, the set of drug targets that are considered can change from drug to drug.

predicted to cause a side effect inflates the number of false positives. We therefore clustered predictions so that a verified prediction for one of the cluster's proteins also explains the predictions for the other cluster members. Proteins predicted to cause the same side effect are grouped into a cluster if it is likely that a drug, binding one of the proteins, will also bind one of the others (using 50% as probability cutoff, see Materials and methods). After excluding predictions for

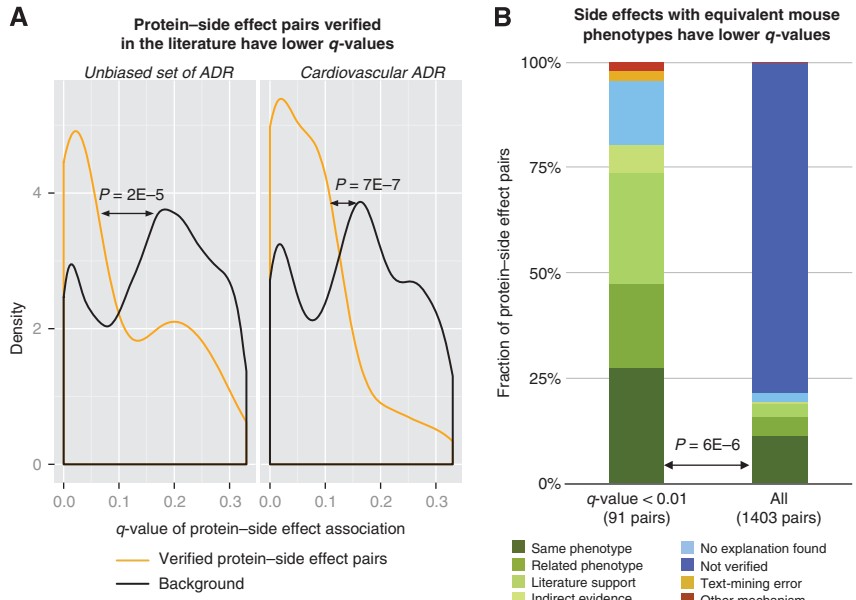

**Figure 2** Validation against independent data sets. (**A**) Two reference sets of protein–side effect pairs were derived from independent sources: 30 cardiovascular-related pairs from Whitebread *et al* (left) and 44 protein–side effect pairs annotated by us from an unbiased survey of the literature by scanning PubMed abstracts for co-occurring proteins and side effects and manually verifying candidates (right, see Supplementary Table 2). A density plot of the *q*-value distribution for protein–side effect associations (i.e., the estimated false discovery rate) shows a significant enrichment in lower *q*-values for pairs within the reference sets. For some proteins of the reference sets, our method predicted whether the protein would be activated or inhibited by drugs causing the side effect. For these, the protein–side effect association with the lowest *q*-value was used. We calculated *P*-values using a one-sided Kolmogorov–Smirnov test. (See Supplementary Figure 10 for a logarithmic version of this plot.) (**B**) Gene–phenotype associations from knockout mice were mapped to human protein–side effect pairs, and related proteins were combined into clusters (see Materials and methods). Of the 91 protein–side effect pairs with a *q*-value of $<0.01$, 25 protein–side effects pairs directly matched phenotypes in mutated mouse strains (27%). This is a significant enrichment over the background rate of 11% (156 exact matches for 1403 protein–side effect clusters, $P = 6 \times 10^{-6}$ using Fisher's exact test; $P = 6 \times 10^{-10}$ without clustering, see Supplementary Figure 4).

metabolizing enzymes, which could cause side effects only indirectly through similar active metabolites for various drugs or through an increased concentration of environmental chemicals, we found 732 side effects to be significantly enriched for at least one drug target (with a *q*-value of $<0.01$, see Supplementary Table 1 and Supplementary Figure 3). For 634 of these side effects, we were able to determine whether they were caused by an inhibition or activation of the candidate target by classifying known drug–target relations by their known modes of action (see Materials and methods). The predicted protein–side effect relations can be accessed at the SIDER database (http://sideeffects.embl.de/).

### Benchmarking against independent data sets

To test if the correlations detected by analyzing package inserts reflected real physiological effects of the targets—caused by drug binding—we benchmarked our data against a number of independent protein–side effect data sets. Known protein–side effect pairs, derived from an unbiased survey of the literature (see Supplementary Table 2) and from a review on cardiovascular-associated side effects (Whitebread *et al*, 2005), have significantly lower *q*-values than the background distribution (Figure 2A). The same is true for protein–side effect pairs derived by mapping gene–phenotype pairs from knockout mice (Figure 2B; Supplementary Figure 4), suggesting that a

deletion of a protein in mice is likely to elicit the same phenotype as inhibiting the respective ortholog in humans despite species and methodology differences. To put our findings into context, we used the set-up of our global benchmark to compare our approach with the performance of the previously proposed 'sparse canonical correlation analysis' method (Mizutani *et al*, 2012). Compared with our predictions, we found a much less significant separation between verified and non-verified predictions (see Supplementary Figure 5).

We manually investigated the most confident causality predictions for each side effect (*q*-value $<10^{-5}$, 116 side effects). Of these, predictions for 72 side effects were directly supported by the literature or mouse phenotypes. Thirteen side effects were more indirectly supported by reports that link the side effect to classes of drugs with a known common mechanism (see Materials and methods). Thus, 73% of the most confident predictions have direct or indirect support from the literature or observed phenotypes (Figure 3). In 22 cases, the link between protein and side effect appeared novel (see Supplementary Table 1). Nine predictions were reported to have other causes, like immunological hypersensitivity reactions (a total of 8% false positives). To be able to extrapolate the accuracy to our large set of 732 predictions (*q*-value $<10^{-2}$), we manually investigated all protein–side effect pairs for which the side effect could be mapped to a mouse phenotype. We find direct support for 80% and a low fraction of false positives (3%, Figure 3). Thus, regarding cases for

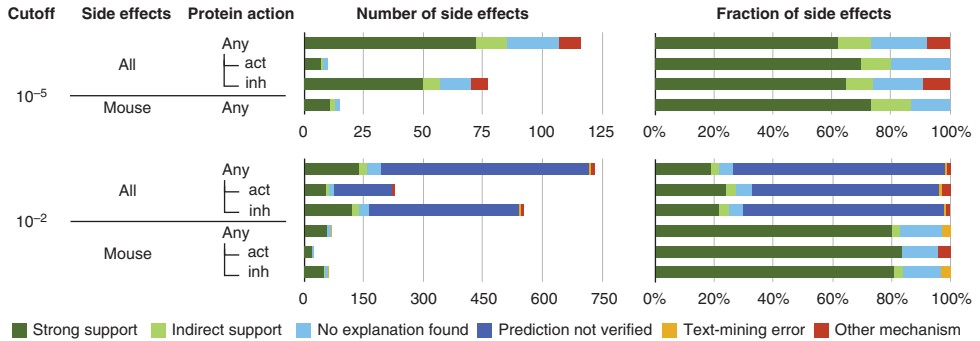

**Figure 3** Benchmarking of the predictions for different false discovery rates. The subset of side effect that corresponds to mouse phenotypes is shown separately. Furthermore, we distinguish if the side effect occurs when protein is activated (act) or inhibited (inh) by drugs. Direct support comes from the existing literature describing the connection between protein and side effect and from mouse phenotypes (dark green). In many cases, the literature suggests multiple causal proteins (medium green). Papers that associate a class of drugs with the same mechanism to a side effect give indirect support (light green). Predictions remained unexplained for two reasons: either not even indirect associations were found in a literature search (light blue), or (for more lenient cutoffs) we did not attempt to verify these predictions (dark blue). In some cases, we found an alternative explanation in the literature (red) or that the association was caused by text-mining errors (e.g., when labels contain generic warnings; yellow).

which we found no or only indirect literature support, 581 of the 732 predictions represent novel associations between individual proteins and a particular side effect that are likely to be causative, often with an implied mode of action (target activation or inhibition, see Supplementary Table 1 and the SIDER database).

These 732 side effects are predicted to be associated with 262 different proteins, corresponding to about three side effects per protein (if each side effect was caused by only one protein). To put this ratio into context, we determined the number of side effects per target in our predictions and in the independent data sets described above (cardiovascular side effects, literature-derived side effects and mouse phenotypes). We found that there are many proteins that cause only one side effect, but a small number of proteins are pleiotropic and elicit many side effects. For all data sets, we find that this distribution follows a power law (Supplementary Figure 6). We conclude from this analysis that the number of predicted side effects per protein is similar to the number observed in the independent data sets.

## *In vivo* validation

To illustrate the power and potential of our large-scale approach, we validated *in vivo* our predicted association between activation of the serotonin receptor 1 family and hyperesthesia (increased pain sensitivity), which is a side effect of triptans, a group of drugs used to treat migraine (*q*-value: $9 \times 10^{-4}$, see also Supplementary Figure 7). This example was selected by a filtering procedure: We first selected side effects predicted to be caused by protein activation (not by inhibition or modulation), that is, to be verifiable in a gene knock-out model. This reduced the number of predictions from 952 to 124. We next excluded 23 clusters of proteins with 5 or more proteins. We finally screened the remaining 101 predictions for those corresponding to a mouse phenotype that was amenable to testing in our experimental setting. Out of the 101 predictions, the tested prediction is the 24th prediction.

Triptans bind to a variety of serotonin receptors, including their main targets, HTR1B and HTR1D, as well as a number of related off-target receptors like HTR7, which had previously been described as an 'HTR1-like receptor' (Hoyer *et al*, 2002). To refine our prediction and to identify the specific target receptor, we gathered additional data on the frequency of hyperesthesia in patients and on the affinities of triptans to their targets (see Supplementary Table 3 for affinities and Supplementary Table 4 for pain-related side-effect frequencies). We found that activation of HTR7 showed the strongest correlation with hyperesthesia (see Supplementary Figure 8). We then tested if the triptan with highest frequency of hyperesthesia in human subjects, zolmitriptan, altered pain sensitivity in mice. In an initial test, we looked for increased mechanosensitivity (von Frey filament test) (Möller *et al*, 1998) and thermal sensitivity (hot plate test) (Mogil *et al*, 1999). While we found no significant effect on mechanosensation (see Supplementary Table 5), zolmitriptan indeed elicits a significant increase in thermal pain sensitivity in mice (Figure 4A). To test whether activation of HTR7 causes the side effect, we pre-treated mice with the selective HTR7 antagonist SB-269970 (Lovell *et al*, 2000). Although SB-269970 had no effect on pain sensitivity alone (Figure 4B), pre-treatment with SB-269970 prevented the effect of zolmitriptan on pain sensitivity (Figure 4C), arguing for a selective blockade of the drug side effect. While *in vitro* tests for cellular activity of zolmitriptan were inconclusive (Supplementary Figure 9) and SB-269970 could influence pain sensation via its reported off-target activity (Kauppila *et al*, 1998; Foong and Bornstein, 2009), a mechanism via HTR7 is consistent with the finding that the research chemical 8-OH-DPAT, an HTR1A and HTR7 agonist (Sprouse *et al*, 2004), elicits hyperesthesia in rats (Millan *et al*, 1989). Furthermore, the triptans with the lowest incidence of hyperesthesia (sumatriptan and naratriptan) have the lowest relative affinity for HTR7 versus the therapeutic target. These data strongly support our prediction that activation of HTR7 by zolmitriptan in patients is responsible for the reported hyperesthesia side effect. Our *in vivo* findings also illustrate the possibility of a directed blocking of a side effect and imply that our systematic computational screen is an

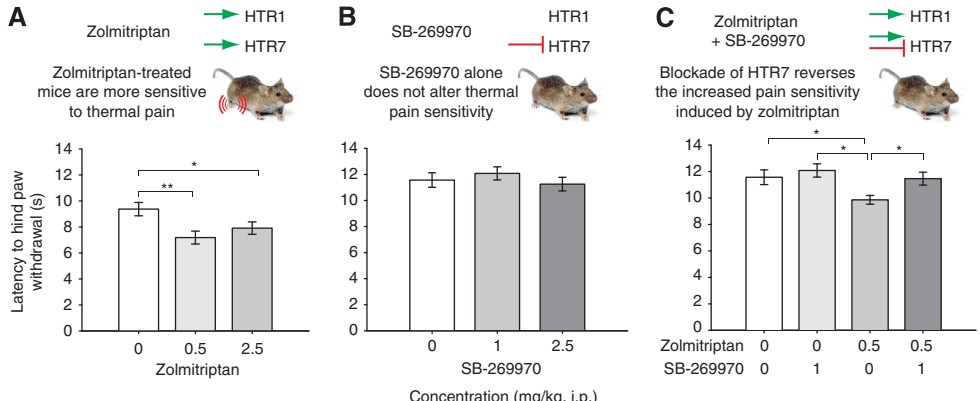

**Figure 4** Activation of HTR7 causes increased pain sensitivity. To detect increased pain sensitivity in mice, the latency to hind paw withdrawal in the hot plate test was measured using an established protocol (Mogil *et al*, 1999). 8-OH-DPAT, a mixed HTR1A/HTR7 agonist (Sprouse *et al*, 2004), was previously found to decrease paw withdrawal latency (Millan *et al*, 1989). (**A**) Zolmitriptan, a broad selectivity serotonin receptor agonist, caused a significant increase in pain sensitivity at both 0.5 and 2.5 mg/kg (i.p.) compared with vehicle-treated controls. (**B**) The HTR7-specific antagonist SB-269970 had no effect on pain sensitivity when delivered alone at concentrations up to 2.5 mg/kg (i.p.). (**C**) Pretreatment of mice with SB-269970 (1.0 mg/kg, i.p.) blocked the hyperalgesic effect of zolmitriptan. Statistical analysis was performed using ANOVA followed by Student–Newman–Keuls *post hoc* test (mean ± s.e.m., significance levels: *$P < 0.05$, **$P < 0.01$; $N = 20$–35). Raw data are shown in Supplementary Table 5.

effective and economical route to the identification of side effect-causing proteins.

## Discussion

The predicted protein–side effect relations can now be used to explain actual drug–side effect pairs that have been recorded in package inserts and to quantify the contributions of various classes of proteins toward the etiology of side effects. To this end, we looked for proteins among the drug's targets that explain an observed side effect among the 732 side effects for which we have identified causal proteins (excluding metabolizing enzymes). For many drugs, there is a target that is thought to mediate the therapeutic effect of the drug. We extracted these drug–target pairs from the literature, and designate these 'canonical targets' as main targets (see Materials and methods). For drugs with known main targets, our method explained 35% of all drug–side effect pairs. We found that main targets are solely responsible for 12% of all drug–side effect pairs (Figure 5A). In 8% of the cases, both main and off-targets elicited the side effect at the same time. Off-targets alone, on the other hand, elicited 15%. The latter category offers the possibility to intervene by reducing off-target activity in analogy to the example described above (Figure 4).

To quantify the influence of major known target families, we subdivided all relevant targets into a number of subsets according to their membership in protein families. Among the five important families of protein targets (Overington *et al*, 2006), we found that G protein-coupled receptors (GPCRs) contributed most to the observed side effects (Figure 5B). As these ratios are dependent on the number of drugs targeting the given targets, we also computed the fraction of explained drug–side effect pairs for subsets of drugs (Figure 5C). Again, the fraction of explained drug–side effect pairs is highest among the drugs that mainly target GPCRs: regardless of on- or off-target activity, 38% of drug–side effect pairs can be

explained by drugs binding to GPCRs. This perhaps reflects the large number of diverse drugs that are available for GPCRs and the high number of known targets for these drugs. GPCRs might mediate side effects more directly than other targets, as they directly influence the nervous system. Phenotypes that are caused by aberrations in the nervous system are readily detected. In contrast, other targets probably cause more subtle intra-organismal phenotypes.

Our approach can only make predictions for proteins that are the targets of a certain number of drugs. If a protein is an off-target of only very few drugs, then there is not enough statistical power to ascribe observed side effects to this protein. Therefore, if a causal connection cannot be made with the current data set, a more complete drug–target matrix may uncover such a connection in the future. However, for very rare off-targets it will remain impossible to statistically detect side effects from clinical data. While more off-targets are discovered, additional targets are unlikely to invalidate existing confident predictions. For example, if a certain side effect always occurs for the main target of a diverse set of drugs, then a newly discovered off-target would need to be connected to all of the drugs to achieve a similar *P*-value. Given that the drug–target matrix is being screened more exhaustively, it seems unlikely to discover such systematic gaps. Nonetheless, very common side effects with multiple etiologies can also not be resolved, as they will not be sufficiently overrepresented for any particular target. Physiologically, combinations of perturbations on different targets will also have different effects than individually perturbing the proteins. However, given the low number of drugs compared with the number of targets and their combinations, detecting differential effects of target combinations is infeasible.

Taken together, for more than half of the investigated side effects, we can predict which proteins cause the side effects upon perturbation by a marketed drug. For the majority of these proteins, we also predict whether their activation or inhibition causes the side effect. Using an animal model, we confirmed that a single protein causes a complex side effect

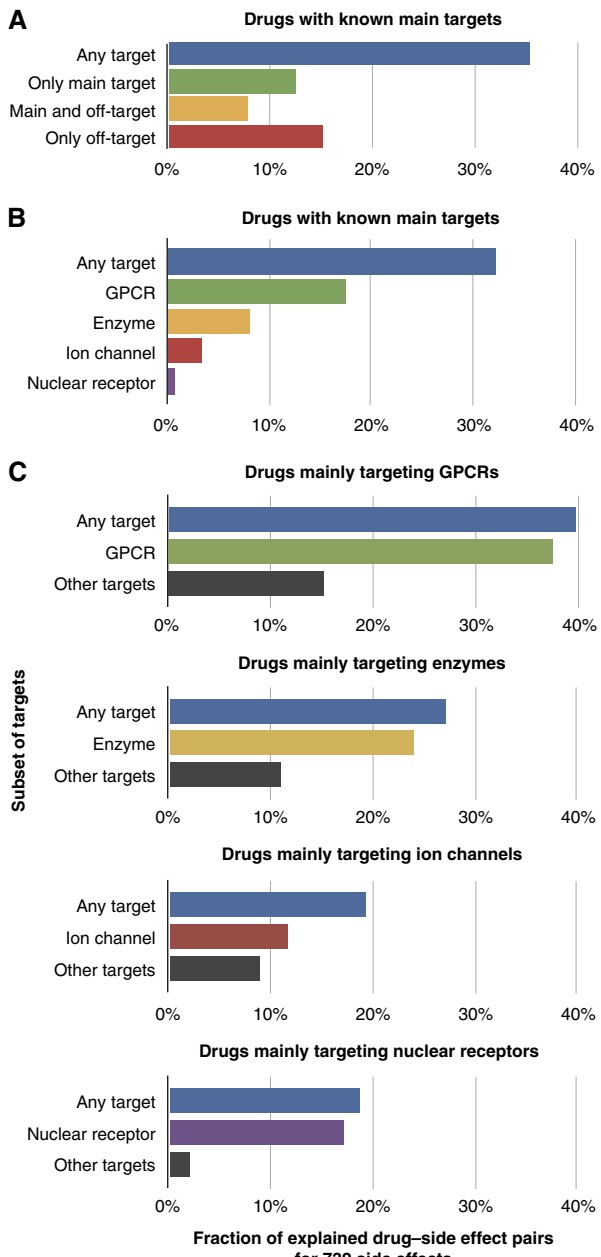

**Figure 5** Drug–side effects pairs explained by main targets and proteins families. (**A**) For each drug, we classify its targets as main targets or off-targets, and check which side effects can be explained by either group (see Figure 1C). Most side effects are mediated through main targets, but there is also a sizeable proportion of drug–side effect pairs where both main and off-targets contribute to the etiology of the side effect. The set of side effects is limited to 732 for which we have identified causal proteins. (**B**) Important families of targets were selected from Overington *et al* (2006), and the fraction of drug–side effect pair that can be explained by the individual protein families is shown. In this figure, 'target' refers to all binding partners that are not metabolizing proteins, and 'enzyme' refers to enzymes that are neither kinases nor metabolizing proteins. (**C**) For each drug, we identify the most prevalent protein family among its main targets, and quantify the contribution of this protein family and the remaining proteins toward the side effects. For example, for drugs that have GPCRs as their main targets, 40% of all drug–side effect pairs can be explained through any target. Almost all of these, namely 38% of all drug–side effect pairs can be explained through the protein subset of GPCRs, which includes both on- and off-target effects. (See Supplementary Figure 11 for the distribution of *q*-values for the different classes of proteins.)

and showed that restoring the altered protein activity with another, specific drug blocks this unwanted complex phenotype. With a more densely populated drug–target matrix that includes all off-targets of a drug, our approach should be able to deduce more protein–phenotype relationships with increased sensitivity and selectivity. Identification of individual proteins that mediate a particular side effect is not only an entry point into the molecular mechanisms underlying side effects, but also opens the possibility of predicting personalized risk of side effects based on large-scale patient genotyping data.

# Materials and methods

## Ethics statement

All animal work has been conducted according to relevant national and international guidelines.

## Drugs and targets

Side effects were imported from the SIDER 2 database (Kuhn *et al*, 2010), which contains information for 996 drugs and 4192 distinct side effects. We import drug–target information from the STITCH 3 database, which in turn is based on the following sources: DrugBank, GLIDA, Matador, PDSP $K_i$ Database, BindingDB, ChEMBLdb and a review article (Roth *et al*, 2000; Imming *et al*, 2006; Okuno *et al*, 2006; Günther *et al*, 2008; Wishart *et al*, 2008; Flockhart, 2009; Gaulton *et al*, 2011). Using a STITCH confidence cutoff of 0.5, we find drug targets for 841 of the drugs with side-effect information.

We curated information on activation and inhibition as different phenotypes occur when a protein is bound by agonists or antagonists. We manually annotated information listed in DrugBank records and Imming *et al* (2006) and added these data to the STITCH 3 database. From STITCH, we also retrieved information derived from Natural Language Processing analysis of PubMed abstracts, Medical Subject Headings (MeSH) pharmacological actions and Anatomical Therapeutic Chemical classification (ATC) entries. If a protein is activated or inhibited, then this annotation is added as a separate virtual protein to the database. For example, a beta blocker will have additional targets representing the inhibited beta-adrenergic receptors. Proteins associated with the same set of drugs are merged into one non-redundant target, as we cannot distinguish them from the drug–target data. For the purpose of calculating significantly enriched side effects, we create a non-redundant set of drugs by removing 218 drugs that are too chemically similar to other drugs using a Tanimoto 2D chemical similarity cutoff of 0.7 as computed with the Chemistry Development Kit (Steinbeck *et al*, 2003) and the redundancy removing algorithm described by Hobohm (Hobohm *et al*, 1992) (using Algorithm 2 of the original paper).

Main targets are meant to be those targets that are commonly thought to mediate the therapeutic effect of a given drug. Of course, so-called off-targets may well also add to the therapeutic mechanism of action in ways that have not been understood yet. We annotated targets as main targets if they were listed in the review by Imming *et al* (2006), in the MeSH Pharmacological Actions and in the ATC codes (if applicable). However, we removed protein families of 10 or more members. Thus, a kinase inhibitor does not have all kinases associated as its main target.

Using the Gene Ontology (GO), targets were classified as GPCRs (GO:0004930), nuclear receptors (GO:0004879), ion channels (GO:0005216), kinases (GO:0016301), or enzymes (GO:0003824, excluding kinases and metabolic enzymes). In case of conflicting annotations for a given protein, associations between the protein and the child GO terms were counted and the top GO term was kept. To classify the main target class of a given drug, again the GO term with the most annotations was kept. For example, a drug annotated as serotonin receptor inhibitor will have both GPCRs and ion channels

(the 5-HT$_3$ receptor family) among its main targets, but GPCRs will usually outnumber the ion channels.

## Prediction of proteins causing side effects

We connect side effects to proteins through the side effect–drug and drug–target relations. For each side effect–protein pair, we count the number of drugs that elicit the side effect, that target the protein, that do both, and that do neither. Then, we calculate the P-value for overrepresentation using Fisher's exact test. Although for over half of all possible combinations between side effects and proteins no drugs are annotated with both the side effect and the protein, 194 560 tested protein–side effect pairs remain to be evaluated. Thus, any P-value needs to be corrected for multiple testing. This is done using the 'qvalue' package for the R programming language (Storey and Tibshirani, 2003). For example, in the analyzed (non-redundant) data, 12 drugs inhibit the adrenergic beta receptor 1. All but of them are annotated to cause bradycardia (decreased heart rate), while among the complete set of 550 drugs, 184 cause this side effect. This yields a P-value of $4 \times 10^{-5}$. Correcting for multiple testing, this still corresponds to a q-value of $9 \times 10^{-4}$.

## Clustering of predictions

Most drugs bind to a wide range of targets. Indeed, it has been shown that drugs will generally bind most members of protein families (Kuhn et al, 2008; Hopkins, 2008), albeit with varying potency. The result of this is that for a given side effect, multiple members of the same protein family will be predicted to cause the side effect. If one of the members of the family is known to cause the side effect, then the other targets should not be regarded as false positives, but rather as a consequence of the broad binding profile of drugs. A similar cause of redundant predictions is the case of protein complexes, where it is not clear which subunit is the actual target. To address this problem, we have implemented a simple clustering mechanism. Each side effect is treated independently. At a given q-value cutoff, there is a number of proteins predicted to be causal for the side effect under consideration. First, proteins known to cause the side effect are identified, for example, by comparison of mouse phenotypes (see below) or by the literature searches. Second, proteins are assigned to the known causal protein, forming clusters of proteins: if at least half of the drugs that bind one target also bind another target that is predicted to cause the same side effect, then we define them to be a cluster.

## Mapping of gene–phenotype associations

For 89 drug targets, we were able to obtain phenotypes of mouse strains carrying null mutations in the target proteins from the Mouse Genome Informatics resource (Bult et al, 2010) and mapped terms from the Mammalian Phenotype Ontology one-to-one to human side effects by strict matching of synonyms, resulting in 116 knockout phenotypes that corresponded to human drug side effects. In total, 398 knockout–phenotype pairs could be mapped to protein–side effect pairs.

## Verification of predictions

Predictions were regarded as directly supported in any of the following cases: The association between protein and side effect has been reported in the literature, or has been observed in a knockout mouse model. A related or similar phenotype was observed in mice that could not be automatically mapped. The opposite phenotype was reported in mutant mice, suggesting an involvement of the protein in the regulation of the observed phenotype despite a different outcome between temporary inhibition and constitutive knockout of the protein. In many cases, the existing literature does not pinpoint a single protein, but rather a group of proteins, to be responsible for a given side effect. Indirect support comes from papers that link a class of drugs with the same mechanism (e.g., selective serotonin reuptake inhibitors) to the side effect.

## Mouse model

Male and female mice of 2 months of age (C57BL/6J and N, Charles River, Calco, Italy and EMMA, Monterotondo, Italy, respectively) were used in this study. Animals were housed in groups of 4–6/cage with food and water ad libitum at 22–24°C on 12-h light cycle with lights on at 0700 h. All procedures were approved by the Italian Ministry of Health and the EMBL Animal Ethics Committee.

## Hot plate test

Thermal sensitivity was measured using the hot plate test (Mogil et al, 1999). Mice were briefly handled each day for 4 days before and on the experimental day. Mice were brought to the testing room 30 min before injection. Testing of female or male mice was alternated. All tests were carried out between 0900 and 1300 h. Mice were placed on a hot plate apparatus (Ugo Basile, Varese, Italy) maintained at $55.0 \pm 0.2°C$ and the latency to the first hind-paw withdrawal was recorded. Animals were removed from the plate after the first response or after 60 s as a cutoff time. Hind-paw withdrawal was scored from videotape (by MAB) as a foot lifting, shake or lick (Hammond, 1993; Espejo and Mir, 1993). Ambulatory movements were not considered as responses.

## Mechanical nociception

Mechanical allodynia was assessed using a Dynamic Plantar Anesthesiometer (Ugo Basile, Varese, Italy) by measuring the latency to withdraw the hind paw from a graded force applied to the plantar surface using a von Frey filament. The electronic von Frey Fiber device applies a single non-flexible filament (0.5 mm in diameter) with increasing force (0.1 g/s; from 0 to 5 g; Wijnvoord et al, 2010) against the plantar surface of the mouse over a 10-s period. The paw withdrawal response automatically turned off the stimulus, and the pressure eliciting the response was recorded. For measurements, mice were placed individually into red enclosures on a framed metal mesh floor and allowed to acclimate for 10 min before testing. Paw withdrawal thresholds were measured in triplicate for each paw of each animal, allowing at least 30 s intervals between successive measurements. In case of no withdrawal up to 5 g this maximum force is maintained until the paw is withdrawn. The paw withdrawal threshold is numerically obtained in seconds (latency of withdrawal). In addition to thermal nociception, we assessed the sensitivity to mechanical nociceptive stimuli using an automated dynamic plantar test (Wijnvoord et al, 2010). No significant effect was observed in the mean paw withdrawal latency for either of dose of Zolmitriptan (0.5 and 2.5 mg/kg, i.p.) used when compared with the saline group. Since no difference was found within the male group or the female group, we combined the male and female data.

## Drugs

Zolmitriptan ((4S)-4-[[3-[2-(Dimethylamino)ethyl]-1H-indol-5-yl]methyl-2-oxazolidinone, Toronto Research Chemicals, Toronto, Canada) and the selective serotonin 7 receptor antagonist SB-269970 ((2-R)-1-[(3-hydroxyphenyl)sulfonyl]-2-[2-(4-methyl-1-piperidinyl)ethyl] pyrrolidine hydrochloride, Sigma-Aldrich, Milan, Italy) were dissolved in saline (0.9% NaCl). Sonication was used to help dissolve zolmitriptan. Mice were injected 30 min before testing with vehicle or zolmitriptan (0.5 and 2.5 mg/kg, i.p.) and, if applicable, pre-treated 20 min prior with vehicle or SB-269970 (1 or 2.5 mg/kg).

## Statistical analysis

Data were analyzed using two-way analysis of variance (ANOVA, sex × treatment). In all cases, no significant effect or interaction involving sex was found. In cases of significance, post hoc testing was performed using Student–Newman–Keuls post hoc test. An α-value of 0.05 was used.

## Supplementary information

## Acknowledgements

We thank Dominika Farley and Raffaele Migliozzi for technical assistance, and acknowledge Martin Hrabe de Angelis and Helmut Fuchs for discussions.

*Author contributions:* MK, MC, LJJ and PB conceived the computational strategy. MK implemented the analysis pipeline and performed the data analysis. MAB and CG designed *in vivo* experiments. MAB supervised and interpreted these experiments. ACG handled the *in vitro* experiments. MK and PB refined the analysis and wrote the paper; all authors edited the manuscript.

## Conflict of interest

The authors declare that they have no conflict of interest.

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
