## [Review Process File · Molecular Systems Biology]

Systematic identification of proteins that elicit drug side effects

Michael Kuhn, Mumna Al Banchaabouchi, Monica Campillos, Lars Juhl Jensen, Cornelius Gross, Anne-Claude Gavin, Peer Bork

Corresponding author: Peer Bork, EMBL

Review timeline:	Submission date:	20 November 2012
	Editorial Decision:	18 December 2012
	Revision received:	31 January 2013
	Accepted:	17 February 2013

Editor: Thomas Lemberger

Transaction Report:

1st Editorial Decision

18 December 2012

Thank you again for submitting your work to Molecular Systems Biology. We have now heard back from two of the three referees who accepted to evaluate the study. Unfortunately, one of the reviewers declared a conflict of interest only after several weeks. Given the present recommendations, I prefer to make a decision now rather than delaying the process. As you will see, the referees find the topic of your study of potential interest and are supportive. They raise however a series of concerns and make suggestions for modifications, which we would ask you to carefully address in a revision of the present work. The recommendations provided by the reviewers are very clear in this regard.

We would kindly ask you to submit 'source data' files (see some instructions at <http://www.nature.com/msb/authors/index.html#a3.4.3>) that provides the measurements (including replicates) depicted in Figure 4A, B, C and in the future supplementary figure that will show the mechanosensitivity results, as required by Reviewer #1.

Referee reports

Reviewer #1 (Remarks to the Author):

In this manuscript, the authors integrate annotation data for drug activities and side effects, based on their STITCH/SIDER-2 database, in order to identify potentially useful cotherapies. They then

validate one of their examples (hyperesthesia due to HTR1 activity) using a mouse pain model.

The STITCH/SIDER-2 database is a valuable resource for drug discovery and side effect prediction, and this work illustrates its power very nicely. The article is clearly and concisely written, and the results provide a nice validation of the approach. As such, I think this manuscript is publishable in MSB. There are other papers out there looking at drug side effects (correctly cited), and this is the best one I've run across so far. I do have a few issues, however, which I would like to see addressed before going to publication. In particular: (1) it is not clear to me how the authors chose their validation example (rank 612 in SupTab.1, and not in SupTab.2), and why the very many higher-ranked side effects were not pursued; and (2) I would like to see a fuller disclosure of the validation data in the supplementary material. (3) I also have minor comments that the authors should consider during revision. More details are given below.

Thus, I would recommend publication of this manuscript in MSB after issue#1 has been satisfactorily explained. I leave it up to the editors to decide the degree to which issue#2 and my more minor issues need to be addressed.

My detailed thoughts and suggestions for improvement on each of the issues are given below.

(1) How was the validation example chosen?

a: It is curious that the authors chose hyperesthesia associated with HTR1 activity, given that this association is ranked #612 in SupTab1, and doesn't even make the cutoffs for SupTab2. This choice suggests that the ranking provided in these supplementary tables, and thus the reasoning behind the described analysis, provides only poor guidance on which side effects could be addressed by cotherapies. I am certain that the picture is not so bleak, so it would be helpful for the authors to more clearly spell out how they came to this example and why they did not pursue one of the many associations that were more highly ranked.

b: If many of the higher-ranked associations were not novel, and it was novelty they sought, this should be explicitly stated as one of the selection rules, and the SupTabs should show their novelty call for each association. If their example is among the top-ranked novel associations, the resulting selection will be more clear.

c: I also think it would be extremely helpful to show the ranking in Fig.4, perhaps as a waterfall plot showing the Q-value on the vertical axis and the rank-ordered side-effect/protein associations on the horizontal axis, coloring each association by its novelty or other key selection category, and indicating which one is the example they pursued.

d: Some statement of how likely it is that they would come up with an example with that association Q-value, given their additional filters for Novelty or other factors would be important here too.

(2) More data on the validation sets should be shown.

a: Given that the authors ran mouse experiments using the filament and heat sensitivity experiments, the results for both should be shown in the supplements.

b: Moreover, all the individual mouse results should be presented in a supplementary table, so that readers would have the option of rederiving the statistical results as they see fit.

c: It is not at all clear to me how the in-vitro validation data were obtained (Fig.S5). The methods should describe this experiment in more detail, and the figure caption should more explicitly explain what we are looking at and what it means.

(3) Minor issues (in order of appearance in manuscript).

a (p.4): Given that many drugs have multiple canonical targets and side effects, I think this methodology assigns greater certainty to those drugs with less pleiotropic annotations. If not, the authors might want to consider such an approach, perhaps by weighting Q-value calculations by $1/\#\text{targets}$.

b (p.7, and Fig.4A): It is interesting that low-dose Zolatriptan is more effective than high-dose. Is this a known trend? If so, it should be cited and commented upon, if not, it might cause concern about the robustness of this side effect prediction. Are there known off-target effects of Zolatriptan which obviate its effectiveness at higher doses?

c (p.8): The authors note that the fraction of explained side-effect-target associations is highest for GPCR targets, but do not even speculate on why this might be the case. Possible explanations that come to mind immediately are that GPCRs are critical for intercellular communication, so it makes perfect sense that they would mediate many known side effects. Have others studied or commented on this question?

d (p.8): It would also be very instructive to show how GPCR targets and the other major target classes fall in the ranking figure that I suggest for issue#1c. I think this could be used to better highlight how importance of different drug classes.

e (Fig.2A): The x-axis really should be $\log_{10}(Q)$ rather than linear Q. The visually obvious differences here are dominated by the lowest significance associations, giving possibly a very misleading impression of the background contribution.

f (Fig.5C): A waterfall plot of the ranked Q-values would be more instructive here, as described above (issue 3C).

g (Fig.5C): Also, is it fair to conclude from this figure that on-target effects account for the majority of reported side-effects, especially for target classes like nuclear receptors? If that is true, this should be commented on in the discussion. If it is really significant and meaningful, this could actually be an important observation about the therapeutic windows available for different target classes, which might deserve more prominence in the manuscript.

Reviewer #2 (Remarks to the Author):

General comments:

The authors propose a method to predict protein-side effect associations from drug-target interaction data and drug-side effect relation data. In vivo validation of a predicted protein-side effect relation is a strong point of this paper. However, there is little discussion about comparison with the previous works with similar objectives.

Major comments:

Associating proteins/pathways with side effects was already proposed in the following papers:

Scheiber et al, *J Chem Inf Model*, 49(2):308-17, 2009.

Xie et al, *PLoS Comput Biol*, 5: e1000387, 2009.

Mizutani et al, *Bioinformatics*, 28, i522-i528, 2012.

The difference of the results/arguments between the previous papers and this paper should be discussed in detail. The 2nd paper is not cited.

In Abstract:

The authors say "732 side effects were predicted to be predominantly caused by individual proteins", but many previous works suggest that side effects are associated with pathways or functional modules rather than individual proteins.

In page 4:

The authors removed target proteins and side effects that are associated with less than five drugs. How did they determine the threshold "five"? Singletons (side effects associated with only one drug) could be removed, but I think that the analysis of rare side effects would be interesting. Why is it not possible to make predictions?

In page 4:

As far as I know, SIDER stores side-effect keywords which are associated with most drugs (e.g., headache, nausea, dizziness, vomiting). Such common side effects are used in the analysis in this paper? The results do not depend on the choice of side effects? In addition, there are some hierarchical relationships between side effect terms. How did the authors handle the problem?

In page 5:

What does it mean by "closely related proteins"? Are they proteins in the same complex or the same pathway? If one subunit in the same complex is known to interact with a drug, other subunits are considered to interact with the drug in the drug-target interaction data in this study? If yes, many proteins tend to be predicted to involve side effects.

In page 5:

The authors finally used only 296 drug targets (about 1% of all human proteins) in the analysis, but is it possible that 732 side effects are associated with such a small number of proteins?

In page 5:

The authors tried to confirm the validity of the predicted protein-side effect associations by the literature search. Another way would be to examine the ability of reconstructing known side effects from proteins using the associations in a cross-validation setting. In this case, it would be interesting to compare the performance with existing side-effect prediction methods (e.g., Atias et al, *Journal of Computational Biology*, 18, 207-218, 2011; Scheiber et al, *J Med Chem*, 52, 3103-3107, 2009).

In page 8:

The method for distinguishing between main targets and off-targets is not clear. What is the "canonical" targets? Is it the original method proposed in this paper?

In page 10:

The authors obtained drug-target interactions from many different sources. However, the definition of drug-target interactions differs from database to database. For example, computationally predicted interactions and experimentally confirmed interactions are mixed, and the difference between direct and indirect interactions is not clear. How did they deal with this problem?

In page 11:

It seems that the proposed method in this paper is conceptually similar to the method in the previous paper (Mizutani et al, *Bioinformatics*, 28, i522-i528, 2012), where protein-side effect pairs were extracted based on the co-occurrence of drugs in protein binding profiles and side-effect profiles and the same side-effect data (SIDER) was used. What is the difference? A comparison should be done.

Figure 1:

It is difficult to understand the figure caption of Figure 1:C. A clear explanation is needed.

Minor comments:

In page 4:

What is the definition of q-value? What is the difference between q-value and p-value? I think that most readers are not familiar with q-value.

In page 7, line 16:

"a mechanisms" should be "a mechanism"?

In page 9, line 22:

"more drug-phenotype" should be "more protein-phenotype"?

In page 12, line 17:

"there is a number of proteins" should be "there are a number of proteins"?

Reviewer #1 (Remarks to the Author):

In this manuscript, the authors integrate annotation data for drug activities and side effects, based on their STITCH/SIDER-2 database, in order to identify potentially useful cotherapies. They then validate one of their examples (hyperesthesia due to HTR1 activity) using a mouse pain model.

The STITCH/SIDER-2 database is a valuable resource for drug discovery and side effect prediction, and this work illustrates its power very nicely. The article is clearly and concisely written, and the results provide a nice validation of the approach. As such, I think this manuscript is publishable in MSB. There are other papers out there looking at drug side effects (correctly cited), and this is the best one I've run across so far. I do have a few issues, however, which I would like to see addressed before going to publication. In particular: (1) it is not clear to me how the authors chose their validation example (rank 612 in SupTab.1, and not in SupTab.2), and why the very many higher-ranked side effects were not pursued; and (2) I would like to see a fuller disclosure of the validation data in the supplementary material. (3) I also have minor comments that the authors should consider during revision. More details are given below.

We would like to thank the reviewer for his positive comments, which we address in detail below.

Thus, I would recommend publication of this manuscript in MSB after issue#1 has been satisfactorily explained. I leave it up to the editors to decide the degree to which issue#2 and my more minor issues need to be addressed.

My detailed thoughts and suggestions for improvement on each of the issues are given below.

(1) How was the validation example chosen?

a: It is curious that the authors chose hyperesthesia associated with HTR1 activity, given that this association is ranked #612 in SupTab1, and doesn't even make the cutoffs for SupTab2. This choice suggests that the ranking provided in these supplementary tables, and thus the reasoning behind the described analysis, provides only poor guidance on which side effects could be addressed by cotherapies. I am certain that the picture is not so bleak, so it would be helpful for the authors to more clearly spell out how they came to this example and why they did not pursue one of the many associations that were more highly ranked.

We agree that the tested example ranks low among all protein–side effect pairs. To pick an example, we only looked at side effects caused by protein activation, so that a mouse knock-out model could potentially be used (going from >900 to 124 predictions). (In the end, we did not use a KO model but a specific inhibitor). We then excluded 23 predictions because more than four proteins we clustered together. We then scanned the list of side effects to those that can be validated in a mouse model, and chose the 24th prediction. Because mouse models are costly, we only tested one prediction.

Action: We have added an explanation on how we chose the phenotype to the “in vivo validation” section.

b: If many of the higher-ranked associations were not novel, and it was novelty they sought, this should be explicitly stated as one of the selection rules, and the SupTabs should show their novelty call for each association. If their example is among the top-ranked novel associations, the resulting selection will be more clear.

As part of our manual annotation of side effects, we have encountered many examples that could not be verified in the current literature. These are the novel associations. Above (and now in the text) we describe the additional criteria that we used to select the validated example.

Action: We have modified the introductory text in Suppl. Table 1 to explain the status of the different predictions and which we consider as novel.

c: I also think it would be extremely helpful to show the ranking in Fig.4, perhaps as a waterfall plot showing the Q-value on the vertical axis and the rank-ordered side-effect/protein associations on the horizontal axis, coloring each association by its novelty or other key selection category, and indicating which one is the example they pursued.

Action: As suggested, we have added a figure to the supplement (Suppl. Fig. 7) to show the relation between rank and q-value more clearly.

d: Some statement of how likely it is that they would come up with an example with that association Q-value, given their additional filters for Novelty or other factors would be important here too.

The more transparent ranking introduced in the text and in Suppl. Fig. 7 should more clearly show the subset of novel associations. We are reluctant to add more statistics on the further filters (activation, available mouse models) as these will be less generally applicable.

(2) More data on the validation sets should be shown.

a: Given that the authors ran mouse experiments using the filament and heat sensitivity experiments, the results for both should be shown in the supplements.

b: Moreover, all the individual mouse results should be presented in a supplementary table, so that readers would have the option of rederiving the statistical results as they see fit.

We apologize for only including summarized data in the first revision of the manuscript.

Action: We have created an additional supplementary table (Suppl. Table 5) that shows the raw data and a summary of the results for the initial filament test.

c: It is not at all clear to me how the in-vitro validation data were obtained (Fig.S5). The methods should describe this experiment in more detail, and the figure caption should more explicitly explain what we are looking at and what it means.

The previous description of the in vitro validation was overly brief, and we have checked the materials of the company that did the tests to provide a better explanation.

Action: We have expanded the caption of this Suppl. Fig. (which is now S9) to include method details and a better explanation.

(3) Minor issues (in order of appearance in manuscript).

a (p.4): Given that many drugs have multiple canonical targets and side effects, I think this methodology assigns greater certainty to those drugs with less pleiotropic annotations. If not, the authors might want to consider such an approach, perhaps by weighting Q-value calculations by $1/\#\text{targets}$.

Certainly, drugs with many targets make the picture more complex. More targets can lead to combinatorial effects that cannot be detected statistically (given the low number of drugs compared to the number of target combinations). However, the q-values are computed for protein–side effect pairs. It turns out that side effects that occur for very many drugs receive lower q-values, because there will be no shared targets between the drugs. In conclusion, we think that our methodology has greater power for drugs with fewer pleiotropic targets, as the reviewer suggests.

b (p.7, and Fig.4A): It is interesting that low-dose Zolatriptan is more effective than high-dose. Is this a known trend? If so, it should be cited and commented upon, if not, it might cause concern about the robustness of this side effect prediction. Are there known off-target effects of Zolatriptan which obviate its effectiveness at higher doses?

The difference between the two Zolmitriptan doses is not significant ($p = 0.305$). Therefore we interpreted the result such that the lower dose already saturates the response. We are not aware of any off-targets that reverse the effect.

c (p.8): The authors note that the fraction of explained side-effect-target associations is highest for GPCR targets, but do not even speculate on why this might be the case. Possible explanations that come to mind immediately are that GPCRs are critical for intercellular communication, so it makes perfect sense that they would mediate many known side effects. Have others studied or commented on this question?

While we did not find other studies on this question, we speculate that the reason is twofold: First, drugs that target GPCRs are well screened for off-targets, and there are relatively many of these drugs. Second, as the reviewer points out, they have more direct effects than intra-cellular targets.

Action: We have added some more speculation on this topic.

d (p.8): It would also be very instructive to show how GPCR targets and the other major target classes fall in the ranking figure that I suggest for issue#1c. I think this could be used to better highlight how importance of different drug classes.

Action: We have added such a plot to the supplement (Suppl. Fig. 11).

e (Fig.2A): The x-axis really should be $\log_{10}(Q)$ rather than linear Q . The visually obvious differences here are dominated by the lowest significance associations, giving possibly a very misleading impression of the background contribution.

We considered both a linear-scale and log-scale version of this plot. While the linear-scale version is dominated by high q -values, there is the opposite problem for the log-scale version: there are some associations with very significant q -values, which in turn dominate the axis. We have therefore chosen to leave the linear-scale version in the manuscript, and add the log-scale version to the supplement.

Action: We have added such a plot to the supplement (Suppl. Fig. 10).

f (Fig.5C): A waterfall plot of the ranked Q -values would be more instructive here, as described above (issue 3C).

As we understand it, the waterfall plot we created to address the issue above should also provide a better illustration for this figure.

g (Fig.5C): Also, is it fair to conclude from this figure that on-target effects account for the majority of reported side-effects, especially for target classes like nuclear receptors? If that is true, this should be commented on in the discussion. If it is really significant and meaningful, this could actually be an important observation about the therapeutic windows available for different target classes, which might deserve more prominence in the manuscript.

We apologize for not making one issue more clear: The same-class effects in Fig. 5C include both on- and off-targets, so we cannot make a more general conclusion here. Stepping back, Fig. 5C suggests that for some classes, many side effects are mediated by targets from other protein families. However, we suspect that the differences between the classes may reflect our lack of knowledge of off-targets in other protein classes.

Action: We have clarified this issue in the figure caption.

Reviewer #2 (Remarks to the Author):

General comments:

The authors propose a method to predict protein-side effect associations from drug-target

interaction data and drug-side effect relation data. In vivo validation of a predicted protein-side effect relation is a strong point of this paper. However, there is little discussion about comparison with the previous works with similar objectives.

Major comments:

Associating proteins/pathways with side effects was already proposed in the following papers: Scheiber et al, J Chem Inf Model, 49(2):308-17, 2009.

Xie et al, PLoS Comput Biol, 5: e1000387, 2009.

Mizutani et al, Bioinformatics, 28, i522-i528, 2012.

The difference of the results/arguments between the previous papers and this paper should be discussed in detail. The 2nd paper is not cited.

We have indeed missed the paper by Xie et al., and thank the reviewer for alerting us to this citation.

Action: we have added a reference to Xie et al., and carved out the differences more in the introduction. A more detailed discussion can be found in the supplement.

In Abstract:

The authors say "732 side effects were predicted to be predominantly caused by individual proteins", but many previous works suggest that side effects are associated with pathways or functional modules rather than individual proteins.

Certainly a target does not cause the side effect in isolation, but in the context of a functional module. Still, to perturb the functional module, the drug needs to bind to an individual protein, which we aim to identify.

Action: we have put a stronger emphasis on the effect of proteins and pathways in the first part of the introduction.

In page 4:

The authors removed target proteins and side effects that are associated with less than five drugs. How did they determine the threshold "five"? Singletons (side effects associated with only one drug) could be removed, but I think that the analysis of rare side effects would be interesting. Why is it not possible to make predictions?

The reason for excluding rare side effects is that we don't have enough statistical power to make reliable predictions. Using a cutoff of five (versus four or six) is an arbitrary choice.

Action: To illustrate the cutoff choice, we have added a figure to the supplement (Suppl. Fig. 1) showing the high number of low-confidence predictions for proteins/side effects with less than five drugs.

In page 4:

As far as I know, SIDER stores side-effect keywords which are associated with most drugs (e.g., headache, nausea, dizziness, vomiting). Such common side effects are used in the analysis in this paper? The results do not depend on the choice of side effects? In addition, there are some hierarchical relationships between side effect terms. How did the authors handle the problem?

Our method intrinsically down-weights frequent and unspecific side effects, so that we did not have to take special care for this problem. These side effects do not correlate well with any target, and so receive only poor q-values (see the new Suppl. Fig. 1).

In our earlier paper on drug–target prediction using side effects, we included hierarchical relationships between the side effects. This increased our prediction efficiency, but also demanded an extra framework of weighting side effects to deal with the redundancy introduced by the hierarchical relationships.

In this work we only used those side effects that are mentioned on the drug labels. This reduces the number of concepts. It is true that there are some terms on the labels that are in a hierarchical relationship. It would be possible to treat these separately, e.g. by combining related side effects. This would have the advantage of making the data set smaller and reducing the problem of multiple hypothesis testing. However, to keep the approach more straightforward we opted to not address the problem of hierarchical side effects.

In page 5:

What does it mean by "closely related proteins"? Are they proteins in the same complex or the same pathway? If one subunit in the same complex is known to interact with a drug, other subunits are considered to interact with the drug in the drug-target interaction data in this study? If yes, many proteins tend to be predicted to involve side effects.

The criterion that we use to define relatedness is that drugs that bind to one protein are likely to also bind the other protein. We called these proteins “closely related” in the previous version of the manuscript, and have now changed this to “pharmacologically similar”. In the case of proteins with similar sequence, the similar binding pocket will lead to promiscuous binding. In the case of complexes like the NMDA receptor, the drug simultaneously binds to the sub-units that are separate entities in our dataset. Proteins from the same pathway are not included (unless they are targets themselves).

If it is known which specific subunit of a complex is bound, then only this specific subunit is annotated as drug target.

Action: We have changed the term in the manuscript to be clearer.

In page 5:

The authors finally used only 296 drug targets (about 1% of all human proteins) in the analysis, but is it possible that 732 side effects are associated with such a small number of proteins?

732 side effects certainly sound like a lot. We use the side effects that are defined in the MedDRA dictionary, and so some side effects will be variations of each other. For the independent datasets and our prediction, we assessed the distribution of side effects per protein. These distributions follow a power law (Suppl. Fig. 6), where there are many proteins that only elicit one side effect, but a few which elicit many side effects. We conclude from this that the number of predicted side effects per protein is in line with the independent datasets.

Action: we have added a supplementary figure, and added a paragraph to the end of the benchmarking section.

In page 5:

The authors tried to confirm the validity of the predicted protein-side effect associations by the literature search. Another way would be to examine the ability of reconstructing known side effects from proteins using the associations in a cross-validation setting. In this case, it would be interesting to compare the performance with existing side-effect prediction methods (e.g., Atias et al, Journal of Computational Biology, 18, 207-218, 2011; Scheiber et al, J Med Chem, 52, 3103-3107, 2009).

Predicting side effect profiles of drugs is a very different research project than the current study. While we identify proteins that have the potential to elicit side effects when perturbed, the side effect will certainly not occur for all drugs that happen to have this target. We thus believe that literature examination is a more direct route to test our predictions.

In page 8:

The method for distinguishing between main targets and off-targets is not clear. What is the "canonical" targets? Is it the original method proposed in this paper?

We apologize for being too brief in the main text. By “canonical targets” we mean those that would be stated in a textbook or in a review article regarding the mechanism of action of the drug, in contrast to the off-targets found by more unbiased screening methods. We are not aware of other

papers that have made this distinction, although it is certainly implicit in works contrasting traditional thinking to poly-pharmacology.

Action: We have made our explanation clearer on how we designate the main targets.

In page 10:

The authors obtained drug-target interactions from many different sources. However, the definition of drug-target interactions differs from database to database. For example, computationally predicted interactions and experimentally confirmed interactions are mixed, and the difference between direct and indirect interactions is not clear. How did they deal with this problem?

This is certainly a problem in all such studies, as the coverage of public K_i data is not sufficient. We rely on STITCH to do this integration, and use a confidence cutoff of 0.5 to filter non-confident interactions (these could be low confidence, or low affinity interactions).

Action: The confidence cutoff is now stated in the methods section.

In page 11:

It seems that the proposed method in this paper is conceptually similar to the method in the previous paper (Mizutani et al, Bioinformatics, 28, i522-i528, 2012), where protein-side effect pairs were extracted based on the co-occurrence of drugs in protein binding profiles and side-effect profiles and the same side-effect data (SIDER) was used. What is the difference? A comparison should be done.

We obtained the data from this paper and extracted protein-side effect predictions. When benchmarked against the three independent data sets, these predictions perform much worse than our predictions (although significantly better than random for 2 of the 3 datasets).

Action: We have added a Suppl. Fig. with these results (Suppl. Fig. 5) and discuss the findings in the benchmarking section.

Figure 1:

It is difficult to understand the figure caption of Figure 1:C. A clear explanation is needed.

Action: We have edited the figure caption.

Minor comments:

In page 4:

What is the definition of q-value? What is the difference between q-value and p-value? I think that most readers are not familiar with q-value.

The formal definition of the q-value is “the minimum false discovery rate at which the test may be called significant.” Thus, the q-value can be treated like a p-value, but it is corrected for multiple hypothesis testing.

Action: We have added the definition to the text.

In page 7, line 16:

"a mechanisms" should be "a mechanism"?

In page 9, line 22:

"more drug-phenotype" should be "more protein-phenotype"?

In page 12, line 17:

"there is a number of proteins" should be "there are a number of proteins"?

Action: Thanks to the referee, we have corrected these errors.